# Pathogenesis of Endometriosis: The Origin of Pain and Subfertility

**DOI:** 10.3390/cells10061381

**Published:** 2021-06-03

**Authors:** Teresa Mira Gruber, Sylvia Mechsner

**Affiliations:** 1Charité–Universitätsmedizin Berlin, Corporate Member of Freie Universität Berlin and Humboldt-Universität zu Berlin, Department of Obstetrics, Augustenburger Platz 1, 13353 Berlin, Germany; teresa-mira.gruber@charite.de; 2Charité–Universitätsmedizin Berlin, Corporate Member of Freie Universität Berlin and Humboldt-Universität zu Berlin, Endometriosis Centre Charité, Department of Gynaecology, Augustenburger Platz 1, 13353 Berlin, Germany

**Keywords:** endometriosis, adenomyosis, pain, infertility, pathophysiology, inflammation, ovarian reserve

## Abstract

Endometriosis (EM) and adenomyosis (AM) are common conditions with pain and infertility as the principal symptoms. The pathophysiology of pain in EM and AM comprises sensory and somatoform pain mechanisms. Over time, these may aggravate and lead to individual complex disease patterns if not diagnosed and treated. Despite the known facts, several years often pass between the onset of symptoms and diagnosis. Chronic pain disorders with changes on a neuronal level frequently arise and are linked to depressive disorders, with the process becoming a vicious cycle. Additionally, women with EM and AM suffer from sub- and infertility. Low fecundity rates are caused by anatomical changes in combination with behavioral changes in the sexual activity of women with chronic pain as well as local proinflammatory factors that not only decrease implantation rates but also promote early abortions.

## 1. Pathogenesis of Endometriotic Lesions

EM is defined by endometrial tissue-like lesions that occur outside the uterine cavity. Primarily, the disease is described as ectopic lesions on the peritoneum of the internal genital organs (endometriosis genitalis externa). However, lately, it has also been characterized as emigration of endometriotic lesions into the myometrium (endometriosis genitalis interna = adenomyosis uteri). The high coincidence of these two EM subtypes may arise in one common pathogenesis [1,2], which has been discussed since 1927. Sampson postulated [3] that the presence of retrograde menstruation is the cause of this phenomenon. Depleted endometrial cells translocate through the tubes into the abdominal cavity and adhere there. The typical manifestation points of EM lesions, the peritoneum of the fossa ovaricae, the Douglas area, the sacrouterine ligaments, and the apex vesicae, strengthen this idea. Presumably, fluid enriched with cells accumulates in anatomic cavities, and then the cells “adhere and grow”. These lesions are often superficial; nevertheless, in many cases, they infiltrate deeper than 5 mm into the surrounding tissue (so-called deep infiltrating endometriosis (DIE)). Why some lesions grow and infiltrate in such a destructive manner is still unknown. Despite the fact that retrograde menstruation is a known and common phenomenon regularly seen in women during laparoscopic procedures, only 10–15% of all women develop EM during their reproductive phase of life [3]. Therefore, other factors such as the type of cells that are translocated into the abdominal cavity, the immune system, genetic and epigenetic processes, as well as environmental factors play a role [4,5]. The occurrence of primary umbilical EM is explained by the reactivation of the coeloma residuals that remain in the navel as a result of the physiological omphaloceles. In the presence of hormones, they differentiate and eventually form EM lesions. Furthermore, the peritoneum itself may favor the adherence of translocated cells and the development of lesions [6]. It is highly likely an immunological tolerance develops towards EM lesions, although the lesions are surrounded by an immunological reaction with inflammation [7]. Several inflammatory markers are increased in the peritoneal fluid of women with EM [8]. A complex interplay of hormonal and immunological factors arises, consisting mainly of an increase in estradiol levels, and is triggered by an increase in prostaglandin E2 levels and an upregulation of the local aromatase. Moreover, macrophages attracted by the secretion of heme create a pathological environment that may favor not only EM but also precancerous and cancerous lesions [9]. The persistent stimulation of immunological factors could give rise to autoimmune disease. These are common in women with EM, yet causality has not yet been established [10]. The distribution of peritoneal EM lesions in the abdominal cavity follows the circular flow of the peritoneal fluid. Lesions are more commonly found in the area of the right paracolic recesses and diaphragm cupola than on the left side. DIE, which involves the intestine, seems to develop as slap-off lesions because these almost always affect the areas of the intestine that have potential contact with the inner genital organs, while the colon (ascending/descending) or the transverse colon are rarely affected [11]. The “tissue injury and repair theory”, by G. Leyendecker, describes the uterus as the origin of the disease. In this theory, uterine hyperperistalsis causes micro traumatization in the junctional zone (JZ) between the endometrium and the myometrium [12,13]. In consequence, proinflammatory mediators are released and induce additional aromatase expression. The then locally released estrogen promotes proliferation and angiogenesis. This leads to changes within the JZ, seen sonographically as an echo-poor hem (halo phenomenon), which represents the attachment of the endometrium. In 3D ultrasounds, the border of the endometrium and myometrium becomes broader and illustrates the transgression of endometrium cells into the myometrium [14,15]. Locally released oxytocin increases myometrial peristalsis and thus initiates a cycle that leads to increasing destruction of the JZ. Presumably, within the processes of mechanical alteration and wound healing, stem cells are activated, which then leave their niche and either enter the abdominal cavity through retrograde menstruation and cause EM or infiltrate into the myometrium and lead to AM [16,17]. In this context of endometrial-myometrial interface disruption (EMID), the upregulation of HIF-1a is likely. HIF-1a then triggers hypoxia-related molecular biological mechanisms that are involved in the establishment of EM lesions and extensive immunological changes occur [18]. It remains unclear whether the immune system primarily has a defect and cannot break down the foreign cells or if those secondarily change the immune system. What we know is that extensive inflammatory reactions and multiple immunological changes are detectable in both the peritoneum and the peritoneal fluid in women with EM [18]. These immunological findings are strongly associated with the occurrence of corresponding lesions, and chronic inflammation plays an increasing role within the pathophysiological theories. Notably, ectopic EM lesions, no matter where they are located (peritoneal lesion, endometrioma at the ovaries, DIE, or in extragenital manifestation in the navel, the abdominal wall, or in the groin) consist not only of epithelial and stromal cells but also of smooth muscle cells. They all express oxytocin and vasopressin receptors as well as estrogen and progesterone receptors [19,20]. Therefore, these lesions are not only endometrial-like settlements but miniature uteri. The smooth muscle cells are in various stages of differentiation and the muscle metaplasia process underlines the origin of lesions from pluripotent cells [19]. Within menstrual blood, cells with characteristics of mesenchymal stem cells have been isolated. These show the potential of differentiation into mesodermal, endodermal, and ectodermal cells, which can explain the existence of ectopic “miniature uteri” [21]. The description of EM cases in males demonstrates another interesting fact about the complexity of the disease origin [22]. The process of fibroblasts within the myofibroblast transformation (FMT), associated with collagen I release, allows the lesions to format further [23]. Endometriotic lesions are always associated with surrounding fibrotic changes. It remains unclear if the surrounding tissue or the lesions themselves trigger these changes. Another theory of Sampson was the distribution of cells via the uterine drainage system (lymphatic and venous) [3]. For many years, there were no new findings, until lymph nodes with EM were detected during bowel surgery [24]. A sentinel lymph node study in rectovaginal EM then demonstrated the presence of EM in marked lymph nodes [25]. Since the occurrence of nodal lesions correlated with the size of the primary rectovaginal lesions as well as demonstrating lymphangiogenesis, a lymphogenic pathway is likely to be involved in the distribution of EM. While the clinical significance of nodal EM is not yet clear, it may reflect an immunological deficit and shows that EM is a systemic disease. Eutopic and ectopic endometrium cell lines were analyzed and differences in receptor expression and sensitivity were described. Ectopic endometrium cells, for example, show resistance to apoptosis by an increased expression of anti-apoptotic proteins such as Bcl-2 and Bcl-xL [26]. Lately, genetics and epigenetics have become an increasingly important focus of scientific interest [2]. Scientists want to estimate the risk of disease, to understand its pathophysiology, and discover new treatment options. EM is heritable in about 50% of cases, as demonstrated in previous twin studies [27,28]. An initial theory was that there could be a major gene that would explain a familial risk for EM. Investigations such as family-based linkage studies have been performed [29]. However, the existence of disease-determining mutations, such as BRCA mutations, seems questionable. Nevertheless, whole genome sequencing studies have managed to detect several significant loci, which have yet to be more closely analyzed [30]. Genetic correlations have been discovered in common gynecological conditions such as uterine leiomyoma and EM. Parallel findings in these two entities may help to understand the underlying biology [31]. Further topics of current interest are epigenetic mechanisms such as DNA methylation, histone coding, or microRNA [32]. Dietary and environmental factors may determine disease onset and progression. For example, dioxin, an omnipresent pollutant, is described as interfering with estrogen signaling through epigenetic modulation [33]. The first experimental data in animal models show an alteration in disease progression by epigenetic modulation, a fact that could be used as a new treatment approach [34,35]. Another still-experimental aspect is the analysis of EM-specific exosomes. Exosomes are extracellular vesicles that carry proteins, lipids, mRNA, microRNA, and DNA. In the peritoneal fluid exosomes with EM, specific proteins have been detected. As in other diseases and malignancies, EM-specific exosomes could pave the way for disease progression [36]. The clinical impact of these findings is not yet clear and needs to be further evaluated.

## 2. Pathophysiologic Origin of Pain

EM is a chronic disease. It recurs after the surgical removal of EM lesions in high numbers and leads to long-term treatment needs in 50% of affected women [37]. Chronic pain and infertility are central problems of our patients. To understand the source of pain that women with EM/AM experience, it is important to understand the pathophysiologic origin of the lesions as well as their effect on the surrounding tissue (Figure 1).

### 2.1. EM-Associated Pain

Typical complaints caused by EM and AM are dysmenorrhea, cyclic and acyclic lower abdominal pain, cyclic dysuria, dyschezia, dyspareunia, as well as infertility. The disease is diagnosed on average 10 years after the onset of the symptoms. Non-specific complaints may lead to consultations of various medical disciplines. Seeing a generalist or another specialist is known to prolong the diagnosis even further. Notably, more than half of women diagnosed with the disease were previously told there was nothing wrong with them [38]. Why are the complaints so difficult to assess? In addition, why should an early diagnosis not be feasible? After all, more than 60% of those diagnosed with EM report that their complaints started before the age of 20 [39]. There is a clear correlation between the duration, the intensity of the complaints, and the extent of the EM and AM manifestations [39]. The treatment of severe EM and AM cases includes challenges: highly qualified gynecological and abdominal surgery, the timing of fertility treatment, and relief of chronic pain. Knowledge of the nature and distribution of EM lesions allows a better understanding of possible effects and the formation of a multidisciplinary treatment approach. In general, all lesions can cause a variety of symptoms. Complaints usually appear in combination, with isolated symptoms being rather rare. The combination of cyclic lower abdominal pain/dysmenorrhea and dyspareunia is typical. Depending on where the lesions are located, somatic (peritoneum, pelvic wall) or visceral (uterus, bladder, or intestine) pain occurs. These two pain characteristics differ. Somatic pain is rather sharp and point-shaped and due to the high density of sensory nerve fibers in the parietal peritoneum, it can be located quite specifically by the person in pain. Visceral pain, on the other hand, is dull and spasm-like. Visceral organs interact with each other, and bladder-induced pain can be hardly distinguished from uterine-induced pain. Besides, the autonomous, visceral innervation interacts with the visceral sensory neurons that pass through the autonomous ganglia. In severe pain, vegetative reactions such as nausea, vomiting, collapse tendency, and cyclic menstrual-associated diarrhea are common complaints in EM/AM patients [40].

### 2.2. Principles of Pain Development

For the perception of pain, a biochemical signal (1) is converted into a neural signal (2) (sensitization of sensory nerve fibers via activation of the nociceptors). At the spinal level, this signal is modulated (3) and referred (attenuated/amplified) to the brain, where the pain perception is occurring (4). Steps one and two are called peripheral sensitization, three and four central sensitizations; disorders of pain perception can occur at all levels and form a complex entity.

### 2.3. Pathogenesis of Specific Forms of Pain

Dysmenorrhea and cyclic lower abdominal pain caused by AM and peritoneal lesions can initially be understood as nociceptive inflammatory pain. Relevant elevated levels of proinflammatory factors such as interleukin-6, -8, TNF-α, and PGE2 have been demonstrated in women with EM [41]. There is a cyclic release of pain and inflammatory mediators. These activate visceral and peritoneal nerve fibers, which lead to an increase in pain sensitivity. Inflammation and cell damage cause the pain and it disappears as the reaction subsides. This form of pain is well managed with non-steroidal anti-inflammatory drugs (NSAIDs) as these cyclooxygenase inhibitors decrease the levels of prostaglandins. Moreover, with the initiation of hormonal therapy, the cycle-related release of mediators is arrested. In some cases, the pain completely disappears. The decrease of cyclic pelvic pain may also be seen in women taking oral contraceptives continuously [42]. In contrast, in AM-related dysmenorrhea, even withdrawal bleeding under hormonal therapy such as combined oral contraceptives (COC) in a cyclical mode can cause severe pain associated with vegetative symptoms such as diarrhea, nausea, and even vomiting. The mechanisms are not yet well understood but it is likely that the primary disorders of the uterine layers with hyperperistalsis result in the release of pain mediators and thus in the activation of pain fibers. Persistent painful withdrawal bleeding under OC must be considered a warning sign. If the disease progresses and DIE (with vaginal, intestinal, or bladder infiltration) develops, other cyclical symptoms will also occur over time. In the case of rectovaginal EM, dyschezia may occur due to the proximity to the intestine or bowel infiltration. Caused by the cyclical swelling of the foci, there may also be cramp-like pain before a bowel movement, stool irregularities, and even cases of cyclical subileus are known. Constipation followed by diarrhea, paradoxical, or even pencil stools may occur. Looking for these specific symptoms will help to identify potential stenosis. It can also affect the rectum, sigmoid, or even the caecal pole region. If EM infiltrates the entire intestinal wall, women describe cyclical hematochezia. Due to the localization of the lesion, acyclical dyspareunia is common, hence EM lesions are known to be hyperinnervated and painful when pressure is applied [43]. EM of the bladder typically leads to cyclical dysuria but may also cause unspecific symptoms such as pollakisuria and pain after voiding the bladder. Cyclical hematuria only occurs if the bladder wall is completely infiltrated, and the urethra is affected.

### 2.4. Neurogenic Inflammation

The most important differential diagnoses of chronic lower abdominal pains next to EM and AM are postoperative adhesions, interstitial cystitis, and non-specific intestinal dysfunction (the irritable colon). These should be considered during clinical evaluation. Furthermore, there is not necessarily a correlation between the extent of EM lesions and pain intensity [40]. Therefore—and this certainly is a phenomenon difficult to understand for physicians—there are only “inconspicuous” findings during the medical examinations. Many patients claim to be in severe pain, while conversely, patients with complex EM and AM may be relatively free of pain. Some patients develop acyclic lower abdominal pain during hormonal therapy (with and without therapeutic amenorrhea). This is an important indication that EM and AM develop mechanisms that can be activated independently of hormonal stimuli. Extensive analyses have been performed regarding the innervation of lesions [44]. Peritoneal lesions show a hyperinnervation of sensory nerve fibers but a loss of sympathetic nerve fibers. It has been demonstrated that the expression of nerve growth factor (NGF) in the peritoneum of women with EM is elevated in comparison to the peritoneum of women without EM [45]. In an analogy to rheumatism research, an imbalance in the release of proinflammatory and anti-inflammatory sympathetic neurotransmitters seems to occur. This imbalance may result in neurogenic inflammation and in causal acyclic pain. Therefore, especially in hormone-therapy-resistant pain, it is important to adjust therapy decisions accordingly. A further complicating factor is the already mentioned adhesion-related pain, which may have both somatic and visceral qualities. Due to chronic pain, patients may develop reactive depression and somatoform pain disorders, which make the clinical picture appear even more complex.

### 2.5. Development of Central Sensitization with Spinal Hyperalgesia

Physiologically, pain is a warning signal. Pain is an individual event, and the perception of pain is subjective. If severe dysmenorrhea (menstrual pain that leads to the need for bedrest or incapacity to go to school or to work without the use of analgesics) remains untreated, it will recur monthly. The pain is initially perceived cyclically, which subsides as the release of the inflammatory and pain mediators decreases. If this pain occurs repeatedly, however, the body’s warning signals take effect, and it is classified as threatening. Now, the modulation at the spinal level does not regulate it down but rather increases it. The release of neurotransmitters is altered (glutamate upregulation), and several modulating mechanisms are set in motion: the nociceptive field is expanded and dysuria and/or dyschezia may occur [46]. These processes lead to spinal hyperalgesia marked by a lowered pain threshold and the perception of pain, even with slight stimuli such as touch. Increasing pain frightens the person experiencing it and makes pain processing more difficult. Severe cramps, accompanied by vegetative reactions, lead the patient to adopt a relieving posture, which is used to seek pain relief. Reactively, this leads to a reflex contraction of the pelvic floor muscles and eventually to pelvic floor dysfunction. This increases the experienced pain and is known to lead to dyspareunia [47]. If these tensions persist, dyspareunia intensifies. Fear of pain during intercourse can strongly influence the ability to relax and a disorder manifests itself. Figure 2 illustrates the process of spinal hyperalgesia. Changes at the central level develop. Functional MRI assessments demonstrated the first morphological adjustments of the brain after a pain latency of two years [48]. Patients suffering from chronic pain have an increased risk of developing complex chronic pain syndromes with bladder dysfunction, irritable bowel syndrome, and vulvodynia [46]. This explains the often severe pain that accompanies patients, even in the absence of pathological findings. Patients are increasingly desperate and look for advice and help but are rarely understood [49]. It is essential to offer pain relief therapy. Pain medication to relieve nociceptive pains (mostly NSAIDs) is widely used. In complex cases, a more differentiated approach is needed, including the use of opioids for severe pain as well as medication that modulates neurotransmitter levels (e.g., gabapentin and pregabalin) in patients with chronic pain disorders. Moreover, physiotherapy or psychosomatic therapy may help in the search for pain relief. Taken together, the pathogenesis of EM-associated pain is very complex and certainly not yet fully understood.

## 3. Pathogenesis of EM-Associated Sub/Infertility

Whether EM/AM as such causes subfertility is a topic of ongoing controversial discussion. It is clear, however, that 25–50% of all infertile patients are affected by EM, especially if they suffer from severe dysmenorrhea and, conversely, 30–50% of all EM patients are subfertile [50]. The fecundity rate of healthy couples is 15–20%, while it is only 2–10% per month in couples in which the woman suffers from untreated EM. Thus, a connection seems to be obvious, but the mechanisms are very complex and not yet understood. For a successful natural conception, the feasibility of sexual intercourse is an important prerequisite. Unfortunately, more than 50% of women with EM complain about dyspareunia that interferes with their sexual activity. Due to the infiltration of the sacrouterine ligaments, women with EM have fewer instances of sexual intercourse, less frequent orgasms, and/or avoid penetration completely. This often puts a strain on couples and makes it more difficult to conceive [51]. The review by Ziegler (2010) [52] provides a very comprehensive overview of the multifactorial genesis of EM/AM and impaired fertility. The uterus itself, but also the ovaries, the peritoneum, and the EM-related inflammatory milieu in the pelvis, alter the rates of successful conception. Chronic inflammation in the peritoneal cavity leads to pronounced adhesions and fundamental anatomical changes and influences physiological function. Therefore, extensive adhesions and anatomical changes are related to impaired fertility.

### 3.1. Peritoneal Lesions and Peritoneal Fluid

The presence of EM in the small pelvis is associated with profound changes in the peritoneal fluid. Peritoneal EM is metabolically active. Compared to normal peritoneal fluid, there are altered concentrations of various cytokines (e.g., IL-6, IL-8, and TNF-α), growth factors (e.g., VEGF), and pain mediators (mainly PGE2) [41]. Angio- and lymphangiogenesis are activated. An increase in the proliferation of mesothelial cells and fibroblasts leads to FMT with myofibroblast formation. The immune system reacts, and immune cells migrate into the tissue and cause an environmental reaction. The proinflammatory EM milieu changes the functionality of the fallopian tubes. Even if the tubes are open in the flow test, the tube factor, which describes the functionality of the tubes, plays a significant role [53]. Moreover, abdominal adhesions and ovarian cysts decrease the chances of spontaneous conception. Patients with rASRM stage III/IV EM show a significantly low pregnancy rate. Notably, the extensive changes within the abdominal cavity not only influence female fertility factors but also have a direct impact on sperm cells. For example, as early as 1996, in vitro experiments showed that EM-associated immune cells and cytokines have a direct effect on sperm mobility and function [54]. It is therefore understandable that even minimal EM (rASRM I/II) may influence fertility rates. In a study by Akande et al., the natural pregnancy rate of women with idiopathic infertility (under 40 years and partners with normal sperm quality) was observed. A significantly lower cumulative pregnancy rate of 36% vs. 55% was found in women with minimal EM [55]. Correspondingly, an improved pregnancy rate was determined after the surgical removal of the EM lesions compared to diagnostic laparoscopy alone [56]. However, since the effect of such interventions only lasts for about one year and then fades, family planning should be considered by the termination of surgery. The benefit does not increase by repeated operations [57].

### 3.2. Endometriomas and the Ovaries

It is frequently discussed whether patients with EM generally show a restriction of ovarian function, or whether this is only the case in the presence of endometrioma. Some studies describe possible changes in ovarian function. For example, reduced preovulatory steroid synthesis and increased levels of inflammatory cytokines indicate altered folliculogenesis and ovulatory dysfunction with reduced oocyte quality [58,59]. Fertility doctors have observed that embryos from women with EM develop more slowly than others [60]. When donor eggs were transferred from women with EM to women without EM, there was a delay in development and a lower implantation rate [61]. These experimental results do not allow generalization, yet the cases included are limited. Further research is needed, but this is the first evidence that women with EM show an altered ovarian function, which may be associated with lower egg quality. Another topic repeatedly discussed is the influence of endometriomas on ovarian function. Women with EM who suffer from infertility show reduced serum levels of AMH [62]. Preliminary operations of endometriomas, and large endometriomas as such, are associated with a decreased follicular reserve. Large endometriomas seem to damage the follicular reserve by pressure on the ovarian cortex as well as by the attraction of proinflammatory immunological mediators. Unfortunately, a reduction of the follicular reserve is almost inevitable during surgical removal of endometriomas. This is not only shown by a reduction of the AMH level, but also by a reduced number of follicles and lower ovulation rates in the operated ovary [63,64]. Therefore, the effects of surgical removal of endometriomas on ovarian function must be weighed against conservative observation. We know that even the removal of small endometriomas causes damage [63]. The presence of endometriomas in one or both ovaries must also be considered. A cut-off of four to five centimeters in diameter of the endometrioma may help to decide for or against surgical treatment [65]. Besides the effects of EM, female age decreases the follicular reserve over time. From the age of 36 years onwards, the effects of surgery become increasingly significant [66]. It is difficult to only look at endometriomas, as they are often associated with higher grade EM and DIE [67]. This is particularly the case if the patients complain about various symptoms (severe dysmenorrhea, intestinal symptoms, dyspareunia, and infertility) [68]. It is also difficult to estimate the isolated influence of endometriomas or DIE on fertility, as these are often nested within a complex clinical picture and are therefore subject to controversy.

### 3.3. AM and Fertility

Finally, it is very important to question to what extent AM influences fertility? Prevalence data are based on histological evidence that can only be reliably determined after hysterectomy. The actual AM prevalence has therefore been underestimated for many years [69]. Sonographically and by MRI, however, more than 80% of all EM patients show signs of AM, and in infertile EM patients, even more than 91% [70]. AM is associated with hyperperistalsis and dysperistalsis, which are accompanied by changes in physiological archimetral function. Hyperperistalsis may have a positive influence on conception in the first years of adulthood and over time it turns into a pathological force. In this context, the evolutionary aspects of EM are important [71]. It has been discussed that in the past young women with good uterine contractility fell pregnant easily, had a better birth outcome, and thus a survival advantage. Since pregnancies and breastfeeding followed each other repeatedly, there was no formation of EM/AM or at least to a lesser extent. Today, however, primigravidas are in their 30s. Thus, women with primary dysmenorrhea and uterine hyperperistalsis are at risk of developing EM/AM. Their uterus has up to two decades to turn an inherently good functionality into a self-destructive force before reproduction is aspired to [72]. A study using uterine contrast medium application showed that directed sperm transport was significantly impaired in women with EM [73]. This may also have consequences for embryo implantation [72]. Besides, local hyperestrogenism may explain the high comorbidity rates of AM and endometrial polyps, endometrial hyperplasia, and myomas that also influence fertility rates [74]. Progesterone is crucial in maintaining the secretory phase of the endometrium and allowing embryo implantation. It has been demonstrated that women with only minor EM have altered steroidogenesis resulting in lower progesterone levels [75]. Progesterone resistance and progesterone receptor downregulation in EM lesions aggravate the local hyperestrogenism and proinflammatory milieu. One further interesting aspect is a possible autoimmune pathway. Experimental data indicate that activated endometrial macrophages have a negative influence on implantation via the release of nitric oxide. Free radicals harm intracellular DNA and cell membranes, destroy fertilized eggs, and inhibit embryo development. The list of further “unhealthy inflammatory” changes with possible effects on fertility is long [76]. Many experimental data suggest inflammation-related limitations, but ultimately the evidence is difficult to establish. It remains unclear whether the endometrium itself changes and leads to disturbed implantation. Endometrial gene expression profiles of patients with AM vs. non-AM patients were established, but no differences were found [77]. However, there were higher rates of miscarriages in the group of AM patients. This was surprising and a first indication that AM affects the course of pregnancy. Notably, even after successful implantation, macrophages and T cells may attack the embryo and lead to early abortions [78]. In another study, patients were classified according to myometrial thickness [79]. The implantation rate, clinical pregnancy rates, and live birth rate were significantly reduced in the group of patients with a myometrial thickness of more than 2.5 cm [79]. Thus, the uterus affected by AM, as the central organ in pregnancy, is increasingly becoming the focus of discussion.

### 3.4. Surgical Treatment and Assisted Reproductive Technology (ART)

The options to increase the chances of conception in women with EM/AM are limited. The existing ESHRE [80], ARSM [65], or NICE [81] guidelines are not consistent and describe differing management approaches. It is necessary to find a compromise between a surgical treatment approach, ART, and/or a combination of both [82]. The treatment plan needs to be individualized and it will depend on the duration of infertility, female age, ovarian reserve, extension, and symptoms of the disease as well as on factors concerning the partner (such as sperm quality). There is scientific evidence on ways to improve the outcome of ART in women with EM/AM-related subfertility [83]. Young patients with severe symptoms, normal physiological ovarian reserve, and good sperm quality may benefit from surgical treatment with complete resection of EM lesions. The operation should take place in an EM-experienced center and be performed diligently. The primary aim will be the restitution of the anatomy in an organ-sparing technique. Postoperative fertility rates of 54% up to 62% have been demonstrated. More than half of these pregnancies had been conceived spontaneously without ART [84,85]. The chances of conceiving naturally are especially high in the first 12–18 months after the operation [86]. Nevertheless, a surgical approach always has to be weighed against the risks: the impairment of ovarian function after the excision of endometriomas, as well as surgical complications, such as infections, thrombosis, embolism, and others. Some of the complications may be long-lasting: for example, neurogenic bladder dysfunction [82]. Therefore, in particular, patients with few symptoms and already reduced ovarian reserve (decrease in AMH level) with endometriomas should be scheduled for immediate ART. The surgical approach should be omitted to avoid decreasing the ovarian function and follicular reserve any further by inevitable surgical damage of the ovaries [86]. Improving ART techniques over the last few years promoted the “freeze-all” strategy. This approach decreases the risk of an ovarian hyperstimulation syndrome. Moreover, EM has been demonstrated to lower the oocyte output during controlled ovarian hyperstimulation [87]. The “freeze-all” approach uses a GnRH antagonist regimen, with GnRH agonist as trigger. After fertilization of the oocytes, the embryos are cryoconserved. They then will be transferred to the uterus one by one in the following cycles. Controlled ovarian stimulation is not necessary. Despite the fact that scientific evidence is still lacking, the “freeze-all” approach may be a promising strategy for women with EM/AM-associated infertility [88]. Notably, controlled ovarian hyperstimulation seems not to affect disease exacerbation [89].

## 4. Summary

EM- and AM-associated pain and subfertility are common conditions among women. There is increasing knowledge about the pathophysiological pathways but both phenomena have not been completely understood yet, and causalities are missing. Due to the complexity of the cases on an individual level, in addition to personal factors, including complaints, age, the extent of EM/AM, and the desire for children, the collective of our patients is extremely heterogeneous. It is our future challenge to conduct meaningful studies to improve therapy options and our patient management.

## Figures and Tables

**Figure 1 cells-10-01381-f001:**
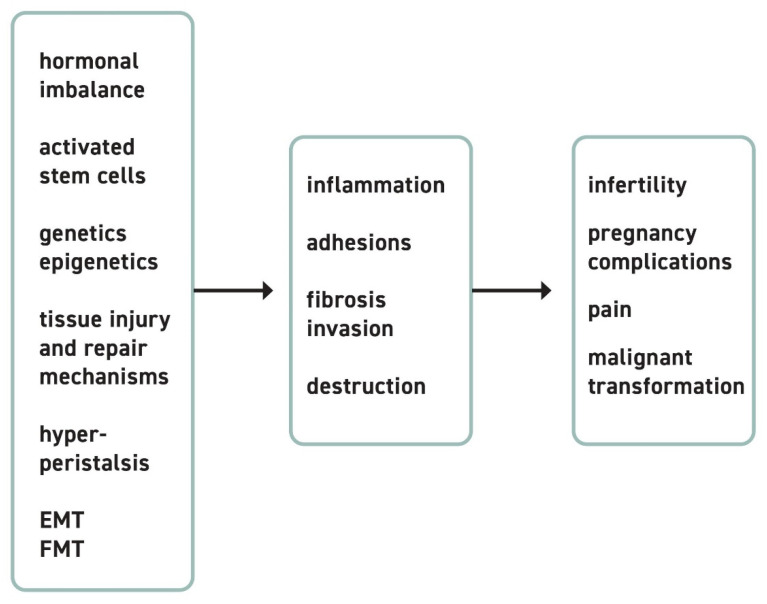
Pathophysiological pathways and clinical consequences. The underlying biological key concepts of EM include hormonal imbalance, activation of stem cells, changes on genetic and epigenetic levels, tissue injury and repair mechanisms, hyperperistalsis, epithelial-mesenchymal transition, and fibroblast to myofibroblast transition. In consequence, the formation of adhesions, fibrosis, inflammation, and local invasion arise. This coexistence of different pathways and their interactions triggers the clinical outcomes that affect multiple organ systems. EM patients present with complaints that include pain, infertility and sterility, complications in pregnancy, and even malignant transformation of preexisting lesions as ovarian cancer.

**Figure 2 cells-10-01381-f002:**
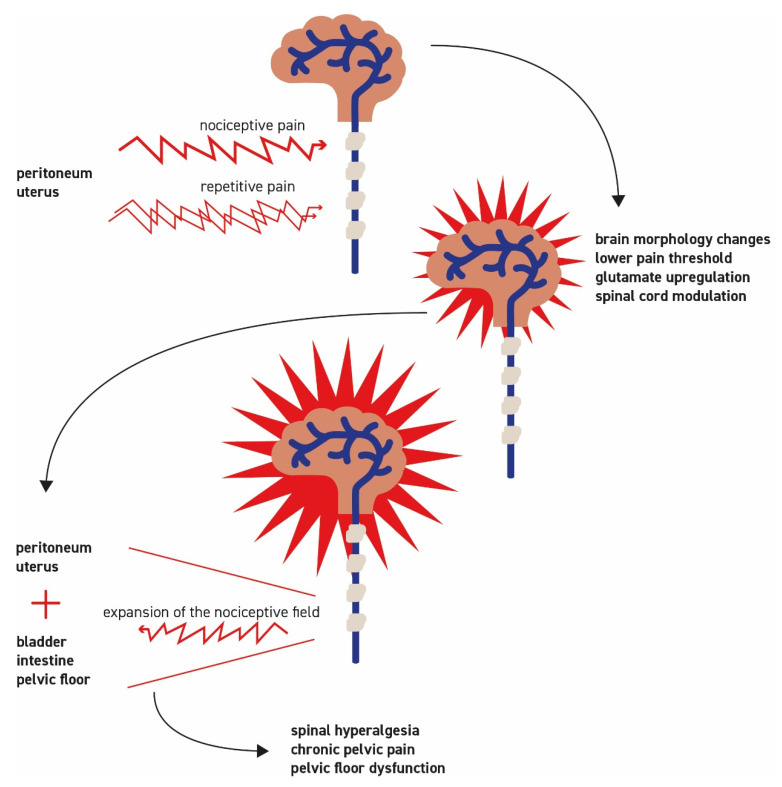
Pathogenesis of spinal hyperalgesia. Patients suffering from chronic pain have an increased risk of developing spinal hyperalgesia, chronic pain syndromes, and pelvic floor dysfunction. Glutamate upregulation and spinal cord modulation occur. After a latency of two years, changes at the central level may develop. This process explains severe pain that accompanies patients, even in the absence of pathological findings.

## Data Availability

Not applicable.

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
