# Peer review of "Pathogenesis of Endometriosis: The Origin of Pain and Subfertility"

_cells, 2021, doi:10.3390/cells10061381_

Round 1

Reviewer 1 Report

Overall: The author of this manuscript provided a brief narrative review on the pathogenesis of endometriosis (and adenomyosis) toward the aspect of pain and infertility. Although this review has provided the most common information on these aspects, however, there is no new knowledge provided to the current research community. Likewise, there is missing a few sections and points which would be critical on providing some useful insights to the readers. Moreover, the comments are shown below:   Abstract: 1) Line 11, the word "adenomyosis" does not need to capitalize. 2) Line 22, the word "endometriosis" and "adenomyosis" can be substituted by "EM" and "AM", respectively.   Section 1: Pathogenesis of endometriotic lesions 1) Beside retrograde menstruation,coelomic metaplasia, oxidative stress and genetic, what about the current theories of autoimmunity and stem cells toward the pathology of endometriosis? 2) What about the changes in multiple immune cells, cytokines, and other factors to promote endometriosis development during the different pathology mentioned? 3) Please add in a section on the current symptoms and management (e.g. hormonal medications, surgery) to provide the readers with more up-to-date information.   Section 2: Pathophysiologic origin of pain 1) Toward inflammation, what about the role of inflammatory cytokines (e.g. TNF, IL-8. etc.) 2) What is the role on peripheral sensitization and central sensitization? 3) What about the mechanisms for the factors proposed? 4)As the increase in neuroangiogenic markers has been linked to a dense nerve supply in lesions and is closely related to pain symptoms in women suffering from endometriosis, this should be mentioned. 5) What about the current treatments and future development for patients with pain?   Section 3: Pathogenesis of endometriosis-associated sub/infertility 1) Please include themechanical, molecular, genetics, and environmental causeson the factors proposed. 2) What about the changes in the progesteroneresistance and dysregulation of progesterone receptorsonimplantation failure? 3) What about the current treatment and future development for patients with infertility?

Author Response

Dear reviewer 1,

Thank you for giving us the opportunity to submit a revised draft of our manuscript “Pathogenesis of endometriosis: the origin of pain and subfertility” for publication in the special issue Cells, Molecular and Cellular Aspects of Endometriosis. We appreciate the time and effort that you and the other reviewers have dedicated to providing your valuable feedback on my manuscript. We are grateful to the reviewers for their insightful comments on our manuscript. We have been able to incorporate changes to reflect most of the suggestions provided by the reviewers. We have highlighted the changes within the manuscript. All page numbers refer to the revised manuscript file. Please see below, in blue, for a point-by-point response to your comments and concerns.

  • The author of this manuscript provided a brief narrative review on the pathogenesis of endometriosis (and adenomyosis) toward the aspect of pain and infertility.

Thank you!

  • Although this review has provided the most common information on these aspects, however, there is no new knowledge provided to the current research community. Likewise, there is missing a few sections and points which would be critical on providing some useful insights to the readers.

While we appreciate the reviewer’s feedback, we respectfully disagree and hope that with the following changes to the manuscript we can convince you otherwise.

  • Moreover, the comments are shown below:   Abstract:1) Line 11, the word "adenomyosis" does not need to capitalize

Thank you for pointing this out. We have corrected the manuscript accordingly (line 11 and the rest of the manuscript).

  • 2) Line 22, the word "endometriosis" and "adenomyosis" can be substituted by "EM" and "AM", respectively.   

Thank you for pointing this out. We have corrected the manuscript accordingly (line 11 and the rest of the manuscript).

  • Section 1: Pathogenesis of endometriotic lesions1) Beside retrograde menstruation, coelomic metaplasia, oxidative stress and genetic, what about the current theories of autoimmunity and stem cells toward the pathology of endometriosis?

Thank you for the comment.  We added the following lines to our manuscript:

“The persistent stimulation of immunological factors could possibly give rise to autoimmune disease. These are common in women with EM, yet causality has not yet been established1“ (line 59-61) and

“Within menstrual blood, cells with characteristics of mesenchymal stem cells have been isolated. These show the potential of differentiation into mesodermal, endodermal, and ectodermal cells that can explain the existence of ectopic ‘miniature uteri’ 2. Evidence that bone marrow cells play a role in disease development has already been established. The description of EM cases in males demonstrates another interesting fact that illustrates the complexity of the disease origin 3.“(line 97-102)

  • 2) What about the changes in multiple immune cells, cytokines, and other factors to promote endometriosis development during the different pathology mentioned?

We think this is an excellent suggestion. We have inserted a new section highlighting the complex interplay (line 50-59):

Furthermore, the peritoneum itself may favour the adherence of translocated cells and the development of lesions 4. It looks like an immunological tolerance develops towards EM lesions, although the lesions are surrounded by an immunological reaction with inflammation 5. Several inflammatory markers are increased in the peritoneal fluid of women with EM 6. A complex interplay of hormonal and immunological factors arises, consisting mainly of an increase in Estradiol levels, and is triggered by an increase in Prostaglandin E2 levels and an upregulation of the local aromatase. Moreover, macrophages attracted by the secretion of heme create a pathological environment that may favor not only EM but also precancerous and cancerous lesions 7.

  • 3) Please add in a section on the current symptoms and management (e.g. hormonal medications, surgery) to provide the readers with more up-to-date information.   

Thank you your comment. We tried to improve the sections on the current symptoms and management:

The section EM-associated pain gives an extensive overview over EM associated symptoms (line 146-172). The management we explain in the different chapters, we point out that in severe cases a multidisciplinary treatment approach is needed (156-160). Specific detail in terms of management you will find in the corresponding sub-sections:

  • Pain (line 186-200)
  • Central sensitization (260-264)
  • Peritoneal lesions and infertility (line 306-310)
  • Endometrioma (line 324-341)
  • We added a new section about surgical treatment and assisted reproductive therapy (Line 386-400)
  • Section 2: Pathophysiologic origin of pain1) Toward inflammation, what about the role of inflammatory cytokines (e.g. TNF, IL-8. etc.)

Thank you for pointing this out. The revised text reads as follows (line 181-185):  
“This form of pain is well manageable with non-steroidal anti-inflammatory drugs (NSAIDs) as these cyclooxygenase inhibitors decrease the levels of prostaglandins. Moreover, with the initiation of hormonal therapy, for example with gestagens (e. g. dienogest 2 mg) at an ovulation inhibition level followed by therapeutic amenorrhea, the cycle-related release of mediators’ arrests. In some cases, the pain completely disappears. The decrease of cyclic pelvic pain may also be seen in women taking oral contraceptives contineusly8. In contrast, in AM-related dysmenorrhea, even withdrawal bleeding under hormonal therapy with combined oral contraceptives (COC) in a cyclical mode can cause severe pain associated with vegetative symptoms as diarrhea, nausea, and even vomiting. The mechanisms are not yet well understood but it is likely that the primary disorders of the uterine layers with hyperperistalsis result in the release of pain mediators and thus in the activation of pain fibers. Note: Persistent strong painful withdrawal bleeding under OC is considered to be a warning! If the disease progresses and DIE (with vaginal, intestinal, or bladder infiltration) develops, other cyclical symptoms will also occur with time”.

  • 2) What is the role on peripheral sensitization and central sensitization?

Thank you for your comment. In the review, section 2.2. (line 173 onwards) gives a detailed overview about the role of peripheral and central sensitization in the pathophysiology of pain and pain disorders. We tried to improve the English language for a better understanding.

  • 3) What about the mechanisms for the factors proposed?

Thank you for pointing this out. Although we think that in chapter 2.4 Neurogenic inflammation (line 213 onwards) the mechanisms are explained. A more detailed level may be beyond the scope of this review.

  • 4) As the increase in neuroangiogenic markers has been linked to a dense nerve supply in lesions and is closely related to pain symptoms in women suffering from endometriosis, this should be mentioned.

Thank you for missing aspect. We inserted an explanation (line 125-131)

“Peritoneal lesions show an hyperinnervation of sensory nerve fibers but a loss of sympathetic nerve fibers. It has be demonstrated that the expression of nerve growth factor (NGF) in the peritoneum of women with EM is elevated in comparison to the peritoneum of women without EM 9. Inan analogy to rheumatism research, an imbalance in the release of pro-inflammatory and anti-inflammatory sympathetic neurotransmitters seems to occur. This imbalance may result in neurogenic inflammation and is causal acyclic pain.”

  • 5) What about the current treatments and future development for patients with pain?   

Thank you for your comment. The current treatment is complex and needs to be adapted to every individual case. We added:  “The treatment of severe EA and AM cases includes challenges: highly qualified gynecological and abdominal surgery, timing of fertility treatment, and relief of chronic pain. Knowledge of the nature and distribution of EM lesions allows a better understanding of possible effects and to form a multidisciplinary treatment approach” (line 156-160).

 We agree that future treatment approaches are an extremely interesting area, nevertheless the analysis is beyond the scope of this review.

  • Section 3: Pathogenesis of endometriosis-associated sub/infertility1) Please include the mechanical, molecular, genetics, and environmental causes on the factors proposed.

Thank you very much for the excellent comment. We tried to reorganise the our text to make this points more clear to the reader.

  1. The central mechanical aspect “uterine hyperpersistalsis” is explained in line 348-359.
    “AM is associated with hyperperistalsis and dysperstialsis, which is accompanied by changes in physiological archimetral function. Hyperperistalsis may have a positive influence on conception in the first years of adulthood and over time it turns into a pathological force. In this context, the evolutionary aspects of EM are important 10. It has been discussed that in the past young women with good uterine contractility fell pregnant easily, had a better birth outcome, and thus a survival advantage. Since pregnancies and breastfeeding followed each other repeatedly, there was no formation of EM/AM or at least to a lesser extent. Today, however, primigravidas are in their 30s. Thus, women with primary dysmenorrhea and uterine hyperperistalsis are at risk of developing archimetrosis followed by EM/AM. Their uterus has up to two decades to turn an inherently good functionality into a self-destructive force before reproduction is aspired. “
  2. Molecular, especially immunological changes in the peritoneal fluid are explained in line 287-29):

“The presence of EM in the small pelvis is associated with profound changes in the peritoneal fluid. Peritoneal EM is metabolically active. Compared to normal peritoneal fluid, there are altered concentrations of various cytokines (e. g. IL-6, IL-8 and TNF-α) , growth factors (e. g. VEGF), and pain mediators (mainly PGE2)11. Angio- and lymphangiogenesis are activated. An increase in the proliferation of mesothelial cells and fibroblasts leads to FMT with myofibroblast formation. The immune system reacts, immune cells migrate into the tissue and cause an environmental reaction. The proinflammatory EM milieu changes the functionality of the fallopian tubes.”

  1. Genetical causes for EM associated subfertility are not yet completely understood. Interestingly the Endometrial gene expression profiles of patients with AM vs. non-AM patients were established, but no differences were found 12 (line 371-372).
  2. Environmental causes for subfertility are numerous. If we agree that anatomical changes are seen as environmental factors Endometriomas, that decrease follicular reserve by pressure on the ovarian cortex over time is of central importance (3.2, line 311 onwards).
  • 2) What about the changes in the progesterone resistance and dysregulation of progesterone receptors on implantation failure?

Thank you for this comment. We added line 364-368:

Progesterone is crucial in maintaining the secretory phase of the endometrium and to allow embryo implantation. It has been shown that women with only minor EM show a shift in steroidogenesis resulting in lower progesterone levels13.  Progesterone resistance and progesterone receptor downregulation in EM lesions aggravate the local hyperestrogenism and pro-inflammatory milieu.

  • 3) What about the current treatment and future development for patients with infertility?

Thank you for this excellent suggestion. We added a section 3.4 to further explain the treatment options on patients suffering from infertility (line 386-400).

3.6. Surgical treatment and assisted reproductive therapy (ART)

“Options to increase the chances of conception in women with EM/AM are limited. No general recommendations or guidelines exist. It is necessary to find a compromise between a surgical treatment approach, ART and/or a combination of both. The treatment plan needs to be individualized and it will depend on the duration of infertility, female age, ovarian reserve, extension, and symptoms of disease as well as on factors concerning the partner (like sperm quality). Young patients with severe symptoms, normal, physiological ovarian reserve, and good sperm quality benefit from extended surgical treatment with complete resection of EM lesions. The primary aim will be the restitution of the anatomy in an organ sparing technique. Postoperativ fertility rates of 54% up to 62% were demonstrated. More than half of these pregnancies had been conceived spontaneously without ART14, 15 . Other patients with little symptoms and already reduced ovarian reserve (decrease in AMH level) with endometriomas should be scheduled for immediate ART. The surgical approach should be omitted to not decrease the ovarian function and follicular reserve any further by an inevitable surgical demage of the ovaries.“

Future treatment innovations may mainly depend on research in the area of ART.

  1. Shigesi, N.; Kvaskoff, M.;  Kirtley, S.;  Feng, Q.;  Fang, H.;  Knight, J. C.;  Missmer, S. A.;  Rahmioglu, N.;  Zondervan, K. T.; Becker, C. M., The association between endometriosis and autoimmune diseases: a systematic review and meta-analysis. Hum Reprod Update 2019, 25 (4), 486-503.
  2. Gargett, C. E.; Schwab, K. E.; Deane, J. A., Endometrial stem/progenitor cells: the first 10 years. Hum Reprod Update 2016, 22 (2), 137-63.
  3. Rei, C.; Williams, T.; Feloney, M., Endometriosis in a Man as a Rare Source of Abdominal Pain: A Case Report and Review of the Literature. Case Rep Obstet Gynecol 2018, 2018, 2083121.
  4. Young, V. J.; Brown, J. K.;  Saunders, P. T.; Horne, A. W., The role of the peritoneum in the pathogenesis of endometriosis. Hum Reprod Update 2013, 19 (5), 558-69.
  5. Scheerer, C.; Bauer, P.;  Chiantera, V.;  Sehouli, J.;  Kaufmann, A.; Mechsner, S., Characterization of endometriosis-associated immune cell infiltrates (EMaICI). Arch Gynecol Obstet 2016, 294 (3), 657-64.
  6. Lin, Y. H.; Chen, Y. H.;  Chang, H. Y.;  Au, H. K.;  Tzeng, C. R.; Huang, Y. H., Chronic Niche Inflammation in Endometriosis-Associated Infertility: Current Understanding and Future Therapeutic Strategies. Int J Mol Sci 2018, 19 (8).
  7. Vercellini, P.; Crosignani, P.;  Somigliana, E.;  Viganò, P.;  Buggio, L.;  Bolis, G.; Fedele, L., The 'incessant menstruation' hypothesis: a mechanistic ovarian cancer model with implications for prevention. Hum Reprod 2011, 26 (9), 2262-73.
  8. Piacenti, I.; Viscardi, M. F.;  Masciullo, L.;  Sangiuliano, C.;  Scaramuzzino, S.;  Piccioni, M. G.;  Muzii, L.;  Benedetti Panici, P.; Porpora, M. G., Dienogest versus continuous oral levonorgestrel/EE in patients with endometriosis: what's the best choice? Gynecol Endocrinol 2021, 1-5.
  9. Arnold, J.; Barcena de Arellano, M. L.;  Rüster, C.;  Vercellino, G. F.;  Chiantera, V.;  Schneider, A.; Mechsner, S., Imbalance between sympathetic and sensory innervation in peritoneal endometriosis. Brain, Behavior, and Immunity 2012, 26 (1), 132-141.
  10. Leyendecker, G., Evolutionäre Aspekte in der Pathogenese und Pathophysiologie von Adenomyose und Endmemtriose. Journal für Gynäkologische Endokrinologie/Österreich 2009, 29, 110-121.
  11. Nanda, A.; K, T.;  Banerjee, P.;  Dutta, M.;  Wangdi, T.;  Sharma, P.;  Chaudhury, K.; Jana, S. K., Cytokines, Angiogenesis, and Extracellular Matrix Degradation are Augmented by Oxidative Stress in Endometriosis. Ann Lab Med 2020, 40 (5), 390-397.
  12. Martínez-Conejero, J. A.; Morgan, M.;  Montesinos, M.;  Fortuño, S.;  Meseguer, M.;  Simón, C.;  Horcajadas, J. A.; Pellicer, A., Adenomyosis does not affect implantation, but is associated with miscarriage in patients undergoing oocyte donation. Fertil Steril 2011, 96 (4), 943-50.
  13. Harlow, C. R.; Cahill, D. J.;  Maile, L. A.;  Talbot, W. M.;  Mears, J.;  Wardle, P. G.; Hull, M. G., Reduced preovulatory granulosa cell steroidogenesis in women with endometriosis. The Journal of Clinical Endocrinology & Metabolism 1996, 81 (1), 426-429.
  14. Dückelmann, A. M.; Taube, E.;  Abesadze, E.;  Chiantera, V.;  Sehouli, J.; Mechsner, S., When and how should peritoneal endometriosis be operated on in order to improve fertility rates and symptoms? The experience and outcomes of nearly 100 cases. Arch Gynecol Obstet 2021.
  15. Abesadze, E.; Sehouli, J.;  Mechsner, S.; Chiantera, V., Possible Role of the Posterior Compartment Peritonectomy, as a Part of the Complex Surgery, Regarding Recurrence Rate, Improvement of Symptoms and Fertility Rate in Patients with Endometriosis, Long-Term Follow-Up. J Minim Invasive Gynecol 2020, 27 (5), 1103-1111.

Reviewer 2 Report

This interesting review discusses the underlying pathogenesis of endometriosis and in part also adenomyosis and the implication for pain perception. The title also entails the word subfertility, but this complex area is covered in much less detail and there is only limited focus on assisted reproduction viewpoints, why my suggestion is that either the reproductive medicine aspects should be removed from this manuscript or substantially deepened and extended. If the latter path is chosen, the manuscript would perhaps benefit from inclusion of an additional expert author with also clinical experience in this area. 

In parts, I found the manuscript provided a stimulating read and in particular the section on pathophysiologic origin of pain was well composed. However, the tissue injury and repair theory of Leyendecker is perhaps insufficiently explained and there is (only?) referencing  to German language publications why I believe this part can be further enhanced by extending this discussion in the context of supportive and/or contrasting viewpoint theories as reflected in other recent review articles. There is – at least in this reviewer’s opinion - a need to expand the pharmacology treatment options where there are many recent advances and moreover, the different types of progestogens are insufficiently covered.

Regrettably, the illustration Figures were difficult to understand and did not fully complement the text. The reason for a photo background of Fig 1 is redundant as most of the text plates are dealing with completely different contexts (‘genetics’, ‘stem cells’, ‘pain’, ‘infertility’ to name but a few) and the positioning of arrows in Fig 2 does not make any sense and I believe the entire graph could be redrawn to better illustrate what is being discussed. 

Throughout the text there are multiple English language misnomers with terminologies and inferences used which are incorrectly or unsuitably applied and I would recommend the authors to consult with a professional editing service to enable a comprehensive and thorough English revision. 

Author Response

Dear reviewer 2,

Thank you for giving us the opportunity to submit a revised draft of our manuscript “Pathogenesis of endometriosis: the origin of pain and subfertility” for publication in the special issue Cells, Molecular and Cellular Aspects of Endometriosis. We appreciate the time and effort that you and the other reviewers have dedicated to providing your valuable feedback on my manuscript. We are grateful to the reviewers for their insightful comments on our manuscript. We have been able to incorporate changes to reflect most of the suggestions provided by the reviewers. We have highlighted the changes within the manuscript. All line numbers refer to the revised manuscript file. Please see below, in blue, for a point-by-point response to your comments and concerns.

  • This interesting review discusses the underlying pathogenesis of endometriosis and in part also adenomyosis and the implication for pain perception.

Thank you very much!

  • The title also entails the word subfertility, but this complex area is covered in much less detail and there is only limited focus on assisted reproduction viewpoints, why my suggestion is that either the reproductive medicine aspects should be removed from this manuscript or substantially deepened and extended. If the latter path is chosen, the manuscript would perhaps benefit from inclusion of an additional expert author with also clinical experience in this area.

While we appreciate the reviewer’s feedback, we respectfully disagree. We think this study makes a valuable contribution to the field. Our work comprises the underlying pathophysiological mechanisms of the two main symptoms in women with EM/AM. We combine the biological facts with medical knowledge, give insights into management and treatment options  

In parts, I found the manuscript provided a stimulating read and in particular the section on pathophysiologic origin of pain was well composed.

Thank you!

  • However, the tissue injury and repair theory of Leyendecker is perhaps insufficiently explained and there is (only?) referencing  to German language publications why I believe this part can be further enhanced by extending this discussion in the context of supportive and/or contrasting viewpoint theories as reflected in other recent review articles.

Thank you for this helpful comment. We extended the first section (line 26-138), and removed the German language references.

  • There is – at least in this reviewer’s opinion - a need to expand the pharmacology treatment options where there are many recent advances and moreover, the different types of progestogens are insufficiently covered.

Thank you for your comment. The authors refer to  Dienogest 2 mg (line 189), our first line treatment in EM/AM. Cyclic or acyclic oral contraceptives may decrease pain levels. Nevertheless, Dienogest has been shown to be the superior 1, 2. We as the authors thinks a more specific review on recent treatment advances is beyond the scope of this review.

  • Regrettably, the illustration Figures were difficult to understand and did not fully complement the text. The reason for a photo background of Fig 1 is redundant as most of the text plates are dealing with completely different contexts (‘genetics’, ‘stem cells’, ‘pain’, ‘infertility’ to name but a few) and the positioning of arrows in

We appreciate your comment, and respectfully disagree. We again discussed the background of the figure and concluded that we find the photo intriguing.
Further we would like to explain the figure to you: the text plates in line three (e. g. hormonal imbalance) describe the underlying biological features, the text plates of the second line describe the pathophysiological consequences (e. g. adhesions) and the ones in the first line describe the symptoms our patients suffer from. The flow is demonstrated by the white arrows. The “syndrome” EM/AM arises by the interplay of all the factors with each other.

  • Fig 2 does not make any sense and I believe the entire graph could be redrawn to better illustrate what is being discussed. 

Thank you for your thoughts. Again, we respectful disagree. The figure demonstrates the interaction of the pelvis and its organ with the central nervous system. Repetitive nociceptive stimuli lead to brain morphological changes. In consequence the nociceptive field is expanded and women with EM/AM will develop pain disorders if not managed well.

  • Throughout the text there are multiple English language misnomers with terminologies and inferences used which are incorrectly or unsuitably applied and I would recommend the authors to consult with a professional editing service to enable a comprehensive and thorough English revision. 

Thank you to point this out. A native-English-speaker reviewed our manuscript and improved the language.

  1. Angioni, S.; Pontis, A.;  Malune, M. E.;  Cela, V.;  Luisi, S.;  Litta, P.;  Vignali, M.; Nappi, L., Is dienogest the best medical treatment for ovarian endometriomas? Results of a multicentric case control study. Gynecol Endocrinol 2020, 36 (1), 84-86.
  2. Piacenti, I.; Viscardi, M. F.;  Masciullo, L.;  Sangiuliano, C.;  Scaramuzzino, S.;  Piccioni, M. G.;  Muzii, L.;  Benedetti Panici, P.; Porpora, M. G., Dienogest versus continuous oral levonorgestrel/EE in patients with endometriosis: what's the best choice? Gynecol Endocrinol 2021, 1-5.

Reviewer 3 Report

The present study aims to review the main evidence regarding the pathogenesis of endometriosis-related pain and subfertility.

The abstract is clear and explicative, the text is well organized in structured paragraphs and the figures proposed are detailed and easily understandable.

A summary table of the selected articles, divided by topic, should be added.

Though, despite the huge and accurate analysis on etiopathogenetic theories, inflammation and central sensitization, there is no mention of environmental factors, pivotal contributors to the onset and development of endometriosis-related pain and infertility. Currently, the leading role of toxic pollutants exposure, dietary factors and oxidative stress represents a central subject of scientific discussion and cannot be omitted.

Moreover, references are often slightly old and might be replaced by more recent evidence.

However, the review deserves publication in Cells, after a thorough major revision.

Comments:

  • Some issues, both in the abstract and in the paper, are weakly expressed and hinder understanding, therefore a revision by a native-English speaker is needed.
  • Abbreviations should only be used after having written the entire word once. Please correct both in the abstract and in the paper.
  • On page 2, line 47, Authors state that the immune system might be involved in the pathogenesis of endometriosis. References should be included. (Porpora MG, et al. High prevalence of autoimmune diseases in women with endometriosis: a case-control study. Gynecol Endocrinol. 2020 Apr;36(4):356-359; Greenbaum H, et al.Evidence for an association between endometriosis, fibromyalgia, and autoimmune diseases. Am J Reprod Immunol. 2019 Apr;81(4):e13095’) Please, add one or more citations.
  • On page 2, line 65, Authors mention Leyendecker’s theory regarding the traumatisms of the junctional zone, potentially involved in the pathogenesis of endometriosis, and cite a German-language article. More relevant evidence by the same Author is available in literature. See Leyendecker G, et al. ‘Endometriosis: a dysfunction and disease of the archimetra’. Hum Reprod Update. 1998 Sep-Oct;4(5):752-62 and Leyendecker G, et al. ‘Adenomyosis and endometriosis. Re-visiting their association and further insights into the mechanisms of auto-traumatisation. An MRI study. Arch Gynecol Obstet. 2015 Apr;291(4):917-32.’ Please replace reference 10.
  • On page 3, Authors discuss several genetic implications in the pathogenesis of endometriosis. However, some references are slightly old and should be replaced by more recent evidence. (Koninckx PR, et al. ‘Pathogenesis of endometriosis: the genetic/epigenetic theory. Fertil Steril. 2019 Feb;111(2):327-340 ; Szukiewicz D, et al. ‘Estrogen- and Progesterone (P4)-Mediated Epigenetic Modifications of Endometrial Stromal Cells (EnSCs) and/or Mesenchymal Stem/Stromal Cells (MSCs) in the Etiopathogenesis of Endometriosis.’ Stem Cell Rev Rep. 2021 Jan 7’.
  • On page 4, in the “Pathogenesis of specific forms of pain”, Authors list several medical approaches to treat endometriosis-related pain and discourage the cyclic administration of combined oral contraceptives (COC) because of the persistent withdrawal bleeding. However, the continuous administration of COC should be mentioned, as their efficacy seems to be almost comparable to that of daily Dienogest 2mg (Piacenti I, et al. ‘Dienogest versus continuous oral levonorgestrel/EE in patients with endometriosis: what's the best choice? Gynecol Endocrinol. 2021 Mar 2:1-5; Angioni S, et al. ‘Is dienogest the best medical treatment for ovarian endometriomas? Results of a multicentric case control study. Gynecol Endocrinol. 2020 Jan;36(1):84-86’.
  • In “Pathogenesis of endometriosis-associated sub/infertility”, several additional factors should be considered. In fact, there is a close link between pain and infertility, consisting of a common origin, due to the anatomical distortions, the presence of adhesions and fibrosis and the immunological disorders. Furthermore, the success rate of assisted reproductive technologies in subfertile/infertile women with reproductive desire should be hinted, as well as the obstetric outcomes of patients achieving pregnancy (higher risk of miscarriage? Higher prevalence of pre-term birth?).

Author Response

Dear reviewer 2,

Thank you for giving us the opportunity to submit a revised draft of our manuscript “Pathogenesis of endometriosis: the origin of pain and subfertility” for publication in the special issue Cells, Molecular and Cellular Aspects of Endometriosis. We appreciate the time and effort that you and the other reviewers have dedicated to providing your valuable feedback on my manuscript. We are grateful to the reviewers for their insightful comments on our manuscript. We have been able to incorporate changes to reflect most of the suggestions provided by the reviewers. We have highlighted the changes within the manuscript. All line numbers refer to the revised manuscript file. Please see below, in blue, for a point-by-point response to your comments and concerns:

  • The present study aims to review the main evidence regarding the pathogenesis of endometriosis-related pain and subfertility.
  • Thank you!
  • The abstract is clear and explicative, the text is well organized in structured paragraphs and the figures proposed are detailed and easily understandable.
  • Thank you!
  • A summary table of the selected articles, divided by topic, should be added.
  • Thank you for the comment. We have been discussion a summary table of articles, but as this is a narrative review the selection of the articles for such a table is not feasible in a comprehensive manner.
  • Though, despite the huge and accurate analysis on etiopathogenetic theories, inflammation and central sensitization, there is no mention of environmental factors, pivotal contributors to the onset and development of endometriosis-related pain and infertility. Currently, the leading role of toxic pollutants exposure, dietary factors and oxidative stress represents a central subject of scientific discussion and cannot be omitted.

Thank you for this missing aspect. We rephrased the section about extrinsic and intrinsic factors modelling disease onset (line 114-139) and added:

“Dietary and environmental factors may determine disease onset and progression. For example, Dioxin, an omnipresent pollutant, is described to interfere with estrogen signalling through epigenetic modulation 1.” (129-131)

  • Moreover, references are often slightly old and might be replaced by more recent evidence. However, the review deserves publication in Cells, after a thorough major revision.

Thank you for your encouraging words!

Comments:

  • Some issues, both in the abstract and in the paper, are weakly expressed and hinder understanding, therefore a revision by a native-English speaker is needed.

Thank you for your comment, we got the manuscript revised by a native-English speaker. We think we were able to improve the language quality.

  • Abbreviations should only be used after having written the entire word once. Please correct both in the abstract and in the paper.

Thanks for the corrections. We changed it in the abstract and the paper.

  • On page 2, line 47, Authors state that the immune system might be involved in the pathogenesis of endometriosis. References should be included. (Porpora MG, et al. High prevalence of autoimmune diseases in women with endometriosis: a case-control study. Gynecol Endocrinol. 2020 Apr;36(4):356-359Greenbaum H, et al.Evidence for an association between endometriosis, fibromyalgia, and autoimmune diseases. Am J Reprod Immunol. 2019 Apr;81(4):e13095’) Please, add one or more citations.

Thanks for the citations, we added them (line 46).

  • On page 2, line 65, Authors mention Leyendecker’s theory regarding the traumatisms of the junctional zone, potentially involved in the pathogenesis of endometriosis, and cite a German-language article. More relevant evidence by the same Author is available in literature. See Leyendecker G, et al. ‘Endometriosis: a dysfunction and disease of the archimetra’. Hum Reprod Update. 1998 Sep-Oct;4(5):752-62 and Leyendecker G, et al. ‘Adenomyosis and endometriosis. Re-visiting their association and further insights into the mechanisms of auto-traumatisation. An MRI study. Arch Gynecol Obstet. 2015 Apr;291(4):917-32.’ Please replace reference 10.

Thanks for the assistance. We replaced reference 10 by your citation (line 69).

  • On page 3, Authors discuss several genetic implications in the pathogenesis of endometriosis. However, some references are slightly old and should be replaced by more recent evidence. (Koninckx PR, et al. ‘Pathogenesis of endometriosis: the genetic/epigenetic theory. Fertil Steril. 2019 Feb;111(2):327-340 ; Szukiewicz D, et al. ‘Estrogen- and Progesterone (P4)-Mediated Epigenetic Modifications of Endometrial Stromal Cells (EnSCs) and/or Mesenchymal Stem/Stromal Cells (MSCs) in the Etiopathogenesis of Endometriosis.’ Stem Cell Rev Rep. 2021 Jan 7’.

Thanks for pointing this out. We rephrased the section and included by the new evidence (line 114-139): “Eutopic and ectopic endometrium cell lines were analysed and differences in receptor expression and sensitivity described. Ectopic endometrium cells, for example, show resistance to apoptosis by an increased expression of anti-apoptotic proteins like Bcl-2 and Bcl-L2. Lately, genetics and epigenetics have become an increasingly important focus of scientific interest3. The aims are to estimate the risk of disease, to understand its pathophysiology, and to discover new treatment options. EM it is heritable in about 50 % of cases as it was demonstrated in previous twin studies 4, 5. An initial theory was that there could be a major gene that explains a familiar risk for EM. Investigations like family-based linkage studies have been performed 6. The existence of disease determining mutations, as for example BRCA mutations, seems questionable. Nevertheless, whole genome sequencing studies have managed to detect several significant loci, which have yet to be more closely analysed 7. Genetic correlations have been discovered in common gynaecological conditions like uterine leiomyoma and EM. Parallel findings in these two entities may help to understand the underlying biology 8. Another topic of current interest are epigenetic mechanisms like DNA methylation, histone coding or MicroRNA9. Dietary and environmental factors may determine disease onset and progression. For example, Dioxin, an omnipresent pollutant, is described to interfere with estrogen signalling through epigenetic modulation 1. The first experimental data in animal models show an alteration in disease progression by epigenetic modulation, a fact that could be used as a new treatment approach 10 11. Another still experimental aspect is the analysis of EM specific exosomes. Exosomes are extracellular vesicles that carry proteins, lipids, mRNA, MicroRNA and DNA. In the peritoneal fluid exosomes with EM specific proteins have been detected. Like in other diseases and malignancies EM specific exosomes could prepare the ground for disease progression 12. The clinical impact of these findings is not yet clear and needs to be further evaluated. “

  • On page 4, in the “Pathogenesis of specific forms of pain”, Authors list several medical approaches to treat endometriosis-related pain and discourage the cyclic administration of combined oral contraceptives (COC) because of the persistent withdrawal bleeding. However, the continuous administration of COC should be mentioned, as their efficacy seems to be almost comparable to that of daily Dienogest 2mg (Piacenti I, et al. ‘Dienogest versus continuous oral levonorgestrel/EE in patients with endometriosis: what's the best choice? Gynecol Endocrinol. 2021 Mar 2:1-5Angioni S, et al. ‘Is dienogest the best medical treatment for ovarian endometriomas? Results of a multicentric case control study. Gynecol Endocrinol. 2020 Jan;36(1):84-86’.

Thank for your comment. We appreciate your feedback, but the authors respectfully disagree. The first line treatment in EM/AM is Dienogest (DNG) 2 mg. This is also supported by the evidence you selected. Piacenti et al. describe a reduction in pain in patients with COC but they state also that there was a significantly higher reduction in endometriotic lesions, pain symptoms, and improvement of the QoL in women taking DNG, than in women taking continuous COC13. Angioni et al compared the administration of cyclic oral progestins (DNE+EE) with continuous administration of DNE 2 mg. They found a similar decrease in pain in both groups but only continuous administration of DNG showed a up to 75% volume reduction in endometriomas during treatment14.

  • In “Pathogenesis of endometriosis-associated sub/infertility”, several additional factors should be considered. In fact, there is a close link between pain and infertility, consisting of a common origin, due to the anatomical distortions, the presence of adhesions and fibrosis and the immunological disorders.

Thank you for your comment. In line 272-279 we explain the association of anatomical distortions and the coexistence of dyspareunia. The consecutive reduction in sexual intercourse makes it more difficult to conceive.

Moreover, we added a section on immunological changes within the abdominal cavity that decrease the function of the fallopian tubes (line 306-310):

The presence of EM in the small pelvis is associated with profound changes in the peritoneal fluid. Peritoneal EM is metabolically active. Compared to normal peritoneal fluid, there are altered concentrations of various cytokines (e. g. IL-6, IL-8 and TNF-α), growth factors (e. g. VEGF), and pain mediators (mainly PGE2)15. Angio- and lymphangiogenesis are activated. An increase in the proliferation of mesothelial cells and fibroblasts leads to FMT with myofibroblast formation. The immune system reacts, immune cells migrate into the tissue and cause an environmental reaction. The proinflammatory EM milieu changes the functionality of the fallopian tubes.

  • Furthermore, the success rate of assisted reproductive technologies in subfertile/infertile women with reproductive desire should be hinted, as well as the obstetric outcomes of patients achieving pregnancy (higher risk of miscarriage? Higher prevalence of pre-term birth?).

Thank you for your helpful comments to improve our review. We added a section 3.4. on surgical therapy and assisted reproductive therapy (ART), line 386-400:

3.4. Surgical treatment and assisted reproductive therapy (ART)

Options to increase the chances of conception in women with EM/AM are limited. No general recommendations or guidelines exist. It is necessary to find a compromise between a surgical treatment approach, ART and/or a combination of both. The treatment plan needs to be individualized and it will depend on the duration of infertility, female age, ovarian reserve, extension, and symptoms of disease as well as on factors concerning the partner (like sperm quality). Young patients with severe symptoms, normal, physiological ovarian reserve, and good sperm quality benefit from extended surgical treatment with complete resection of EM lesions. The primary aim will be the restitution of the anatomy in an organ sparing technique. Postoperative fertility rates of 54% up to 62% were demonstrated. More than half of these pregnancies had been conceived spontaneously without ART16, 17 . Other patients with little symptoms and already reduced ovarian reserve (decrease in AMH level) with endometriomas should be scheduled for immediate ART. The surgical approach should be omitted to not decrease the ovarian function and follicular reserve any further by an inevitable surgical damage of the ovaries.

The obstetric outcomes and complications, we believe, are beyond the scope of this review.

  1. Giampaolino, P.; Della Corte, L.;  Foreste, V.;  Barra, F.;  Ferrero, S.; Bifulco, G., Dioxin and endometriosis: a new possible relation based on epigenetic theory. Gynecological Endocrinology 2020, 36 (4), 279-284.
  2. Delbandi, A. A.; Mahmoudi, M.;  Shervin, A.;  Heidari, S.;  Kolahdouz-Mohammadi, R.; Zarnani, A. H., Evaluation of apoptosis and angiogenesis in ectopic and eutopic stromal cells of patients with endometriosis compared to non-endometriotic controls. BMC Womens Health 2020, 20 (1), 3.
  3. Koninckx, P. R.; Ussia, A.;  Adamyan, L.;  Wattiez, A.;  Gomel, V.; Martin, D. C., Pathogenesis of endometriosis: the genetic/epigenetic theory. Fertil Steril 2019, 111 (2), 327-340.
  4. Treloar, S. A.; O'Connor, D. T.;  O'Connor, V. M.; Martin, N. G., Genetic influences on endometriosis in an Australian twin sample. [email protected]. Fertil Steril 1999, 71 (4), 701-10.
  5. Saha, R.; Pettersson, H. J.;  Svedberg, P.;  Olovsson, M.;  Bergqvist, A.;  Marions, L.;  Tornvall, P.; Kuja-Halkola, R., Heritability of endometriosis. Fertil Steril 2015, 104 (4), 947-952.
  6. Zondervan, K. T.; Treloar, S. A.;  Lin, J.;  Weeks, D. E.;  Nyholt, D. R.;  Mangion, J.;  MacKay, I. J.;  Cardon, L. R.;  Martin, N. G.; Kennedy, S. H., Significant evidence of one or more susceptibility loci for endometriosis with near-Mendelian inheritance on chromosome 7p13–15. Human Reproduction 2007, 22 (3), 717-728.
  7. Sapkota, Y.; Fassbender, A.;  Bowdler, L.;  Fung, J. N.;  Peterse, D.;  O, D.;  Montgomery, G. W.;  Nyholt, D. R.; D'Hooghe, T. M., Independent Replication and Meta-Analysis for Endometriosis Risk Loci. Twin Res Hum Genet 2015, 18 (5), 518-25.
  8. Gallagher, C. S.; Mäkinen, N.;  Harris, H. R.;  Rahmioglu, N.;  Uimari, O.;  Cook, J. P.;  Shigesi, N.;  Ferreira, T.;  Velez-Edwards, D. R.;  Edwards, T. L.;  Mortlock, S.;  Ruhioglu, Z.;  Day, F.;  Becker, C. M.;  Karhunen, V.;  Martikainen, H.;  Järvelin, M. R.;  Cantor, R. M.;  Ridker, P. M.;  Terry, K. L.;  Buring, J. E.;  Gordon, S. D.;  Medland, S. E.;  Montgomery, G. W.;  Nyholt, D. R.;  Hinds, D. A.;  Tung, J. Y.;  Perry, J. R. B.;  Lind, P. A.;  Painter, J. N.;  Martin, N. G.;  Morris, A. P.;  Chasman, D. I.;  Missmer, S. A.;  Zondervan, K. T.; Morton, C. C., Genome-wide association and epidemiological analyses reveal common genetic origins between uterine leiomyomata and endometriosis. Nat Commun 2019, 10 (1), 4857.
  9. Zelenko, Z.; Aghajanova, L.;  Irwin, J. C.; Giudice, L. C., Nuclear receptor, coregulator signaling, and chromatin remodeling pathways suggest involvement of the epigenome in the steroid hormone response of endometrium and abnormalities in endometriosis. Reprod Sci 2012, 19 (2), 152-62.
  10. Zheng, Y.; Khan, Z.;  Zanfagnin, V.;  Correa, L. F.;  Delaney, A. A.; Daftary, G. S., Epigenetic Modulation of Collagen 1A1: Therapeutic Implications in Fibrosis and Endometriosis. Biol Reprod 2016, 94 (4), 87.
  11. Szukiewicz, D.; Stangret, A.;  Ruiz-Ruiz, C.;  Olivares, E. G.;  Soriţău, O.;  Suşman, S.; Szewczyk, G., Estrogen- and Progesterone (P4)-Mediated Epigenetic Modifications of Endometrial Stromal Cells (EnSCs) and/or Mesenchymal Stem/Stromal Cells (MSCs) in the Etiopathogenesis of Endometriosis. Stem Cell Rev Rep 2021.
  12. Nazri, H. M.; Imran, M.;  Fischer, R.;  Heilig, R.;  Manek, S.;  Dragovic, R. A.;  Kessler, B. M.;  Zondervan, K. T.;  Tapmeier, T. T.; Becker, C. M., Characterization of exosomes in peritoneal fluid of endometriosis patients. Fertil Steril 2020, 113 (2), 364-373.e2.
  13. Piacenti, I.; Viscardi, M. F.;  Masciullo, L.;  Sangiuliano, C.;  Scaramuzzino, S.;  Piccioni, M. G.;  Muzii, L.;  Benedetti Panici, P.; Porpora, M. G., Dienogest versus continuous oral levonorgestrel/EE in patients with endometriosis: what's the best choice? Gynecol Endocrinol 2021, 1-5.
  14. Angioni, S.; Pontis, A.;  Malune, M. E.;  Cela, V.;  Luisi, S.;  Litta, P.;  Vignali, M.; Nappi, L., Is dienogest the best medical treatment for ovarian endometriomas? Results of a multicentric case control study. Gynecol Endocrinol 2020, 36 (1), 84-86.
  15. Nanda, A.; K, T.;  Banerjee, P.;  Dutta, M.;  Wangdi, T.;  Sharma, P.;  Chaudhury, K.; Jana, S. K., Cytokines, Angiogenesis, and Extracellular Matrix Degradation are Augmented by Oxidative Stress in Endometriosis. Ann Lab Med 2020, 40 (5), 390-397.
  16. Dückelmann, A. M.; Taube, E.;  Abesadze, E.;  Chiantera, V.;  Sehouli, J.; Mechsner, S., When and how should peritoneal endometriosis be operated on in order to improve fertility rates and symptoms? The experience and outcomes of nearly 100 cases. Arch Gynecol Obstet 2021.
  17. Abesadze, E.; Sehouli, J.;  Mechsner, S.; Chiantera, V., Possible Role of the Posterior Compartment Peritonectomy, as a Part of the Complex Surgery, Regarding Recurrence Rate, Improvement of Symptoms and Fertility Rate in Patients with Endometriosis, Long-Term Follow-Up. J Minim Invasive Gynecol 2020, 27 (5), 1103-1111.

Round 2

Reviewer 1 Report

This review article has been significantly improved and it is ready for publication.

Author Response

Thank you very much

Reviewer 2 Report

The authors are acknowledged for their continued effort of improving this review manuscript. However, using their own terminology, I respectfully disagree with some of the arguments provided. 

The title entails the word subfertility and yet the review fails to recognize many of the recent studies highlighting a possible impact (or not) on the oocyte eg a recent review Simopoulou et al., Biomedicines 2021, 9, 273.

The authors claim the no general guidelines exist (line 385) in EM associated infertility which is completely neglecting referencing to ESHRE, ASRM, NICE guidelines, to mention but a few. Moreover, simply stating that embryos in women with EM develop slower (line 316), providing a reference from almost 30 years ago, which is contradicted by other more recent studies, certainly does not recognize the enormous developments in ART (the T is for technology, not therapy!)  practices during the last decade or so. Moreover, given the estrogen-dependence associated with endometriosis, it is plausible that hyper-stimulation associated with assisted reproduction treatment may exacerbate the disease process and adversely affect endometrial receptivity and subsequent implantation. However, recent large datasets have not been able to demonstrate lower live birth rates in EM patients compared to other indications, when undergoing ART. Perhaps a discussion on this topic would be appropriate in this way, as it has been proposed that a freeze-all deferred ET approach would mitigate these effects and improve IVF outcomes in patients with endometriosis.

I also do not agree with the purported statement that the available medical treatments may be outside the scope of this paper, and I consider this to be a not appropriately balanced viewpoint by the authors to provide the statement that dienogest is the ´superior´ progestogen therapy without giving a more comprehensive and detailed overview of all the available progestogens, their mechanisms of action and different delivery systems available. If such an expansion is considered beyond the scope of this review, please delete the statements related to dienogest, as there are also many side effects associated with this compound which are not mentioned.   

The authors seem to introduce the notion that surgery may still be first line option for the young patient with severe pain and no other (couple dependent) fertility compromising factors. This may then pose a substantial risk if not performed diligently and in fact many recent publications indicate the contrary, that surgery is not the primary choice of therapy and should be reserved for carefully selected cases. Such a more balance view is reflected in the recent reviews in Nature Reviews (Chapron 2019) and NEJM (Taylor 2021) why I think this notion needs to be discussed in more detail. If the author insists, let it reflect this is their opinion and what is the basis for this.

The illustrations have not been redone, which I find unfortunate. A review with clear and supporting illustrations have a much higher educational value and as such I disagree with keeping the photo as background and I still believe the spinal hyperalgesia figure can be improved for better understanding. 

Finally, while the language has improved, the manuscript deserves additional oversight by an English narrator since there are still several grammatical and several spelling errors in addition to replacement of remaining German-only language citation(s).

Author Response

Dear reviewer 2,

Thank you for your thoughts and the effort to improve our manuscript. We appreciate your help. We have highlighted the changes within the manuscript. All page numbers refer to the revised manuscript file. Please see below, in blue, for a point-by-point response to your comments and concerns:

  • The authors are acknowledged for their continued effort of improving this review manuscript.
    Thank you !
  • However, using their own terminology, I respectfully disagree with some of the arguments provided. The title entails the word subfertility and yet the review fails to recognize many of the recent studies highlighting a possible impact (or not) on the oocyte eg a recent review Simopoulou et al., Biomedicines 2021, 9, 273.

The authors claim the no general guidelines exist (line 385) in EM associated infertility which is completely neglecting referencing to ESHRE, ASRM, NICE guidelines, to mention but a few.

Yes, thanks for pointing this out. The guidelines do exist, nevertheless, they differ and don’t have a standard plan. Our patients with various symptoms, disease patterns and treatment expectations deserve an individual treatment approach.
We tried to improve our manuscript line 377-379:
The options to increase the chances of conception in women with EM/AM are limited. The existing ESHRE1, ARSM2 or NICE3 guidelines are not consistent and describe altering management approaches.

  • Moreover, simply stating that embryos in women with EM develop slower (line 316), providing a reference from almost 30 years ago, which is contradicted by other more recent studies, certainly does not recognize the enormous developments in ART (the T is for technology, not therapy!)  practices during the last decade or so.

 Thank you for your comment. We corrected our mistake. The correct term is ART Assisted reproductive technology (line 384). Notably, the recent review of Simopoulou et al, 2021, also cites the same article as we do from 30 years ago to give evidence for the “reduced embryo development capacity and implantation dynamics”4

  • Moreover, given the estrogen-dependence associated with endometriosis, it is plausible that hyper-stimulation associated with assisted reproduction treatment may exacerbate the disease process and adversely affect endometrial receptivity and subsequent implantation. However, recent large datasets have not been able to demonstrate lower live birth rates in EM patients compared to other indications, when undergoing ART. Perhaps a discussion on this topic would be appropriate in this way, as it has been proposed that a freeze-all deferred ET approach would mitigate these effects and improve IVF outcomes in patients with endometriosis.

Thank you for this interesting perspective. We included your ideas in the last paragraph (line 395-407):

Therefore, especially patients with little symptoms and already reduced ovarian reserve (decrease in AMH level) with endometriomas should be scheduled for immediate ART. The surgical approach should be omitted to not decrease the ovarian function and follicular reserve any further by inevitable surgical damage of the ovaries5. Improving ART techniques over the last years promoted the “freeze-all” strategy. This approach decreases the risk of an ovarian hyperstimulation syndrome. Moreover, EM has been demonstrated to lower the oocyte output during controlled ovarian hyperstimulation6. The “freeze-all” approach uses an GnRH antagonist regimen, with GnRH agonist as trigger. After fertilization of the oocytes, the embryos are cryoconserved. They then will be transferred to the uterus one by one in the following cycles. Controlled ovarian stimulation is not necessary. Despite that scientific evidence is still lacking the “freeze-all” approach may be a promising strategy in women with EM/AM associated infertility 7. Notably, controlled ovarian hyperstimulation seem not to affect disease exazerbation.8

  • I also do not agree with the purported statement that the available medical treatments may be outside the scope of this paper, and I consider this to be a not appropriately balanced viewpoint by the authors to provide the statement that dienogest is the ´superior´ progestogen therapy without giving a more comprehensive and detailed overview of all the available progestogens, their mechanisms of action and different delivery systems available. If such an expansion is considered beyond the scope of this review, please delete the statements related to dienogest, as there are also many side effects associated with this compound which are not mentioned.   

Thank you for your comment. We removed the statement (line 186).

  • The authors seem to introduce the notion that surgery may still be first line option for the young patient with severe pain and no other (couple dependent) fertility compromising factors. This may then pose a substantial risk if not performed diligently and in fact many recent publications indicate the contrary, that surgery is not the primary choice of therapy and should be reserved for carefully selected cases. Such a more balance view is reflected in the recent reviews in Nature Reviews (Chapron 2019) and NEJM (Taylor 2021) why I think this notion needs to be discussed in more detail. If the author insists, let it reflect this is their opinion and what is the basis for this.
    Thank you for your comment. It was not our aim to introduce the notion that surgery may still be the first line therapy. And we strongly agree with you that surgery poses a substantial risk if not performed diligently. We tried to rewrite the passage to express that we do recommend a surgical approach in a cohort of well selected patients (line 379-395):

It is necessary to find a compromise between a surgical treatment approach, ART, and/or a combination of both9. The treatment plan needs to be individualized and it will depend on the duration of infertility, female age, ovarian reserve, extension, and symptoms of the disease as well as on factors concerning the partner (like sperm quality). There is scientific evidence on ways to improve the outcome of ART in women with EM/AM-related subfertility10. Young patients with severe symptoms, normal, physiological ovarian reserve, and good sperm quality may benefit from surgical treatment with complete resection of EM lesions. The operation should take place in an EM-experienced center and be performed diligently. The primary aim will the restitution of the anatomy in an organ sparing technique. Postoperative fertility rates of 54% up to 62% were demonstrated. More than half of these pregnancies had been conceived spontaneously without ART11, 12. The chances of conceiving naturally are especially high in the first 12-18 months after the operation5. Nevertheless, a surgical approach always has to be weighed against the risks: the impairment of ovarian function after the excision of Endometriomas, as well as surgical complications, such as infections, thrombosis, embolism, and others. Some of the complications may be long-lasting for example neurogenic bladder dysfunction 9.
The illustrations have not been redone, which I find unfortunate. A review with clear and supporting illustrations have a much higher educational value and as such I disagree with keeping the photo as background and I still believe the spinal hyperalgesia figure can be improved for better understanding.

Thanks again for insisting on improvements. We tried our best to rearrange the figure and are happy with the results.

  • Finally, while the language has improved, the manuscript deserves additional oversight by an English narrator since there are still several grammatical and several spelling errors in addition to replacement of remaining German-only language citation(s).
    Thanks for your critical review, we again handed the manuscript to a native speaker and revised the manuscript accordingly. Moreover, we removed all the German-only language citations.

Literature:

  1. Dunselman, G. A. J.; Vermeulen, N.;  Becker, C.;  Calhaz-Jorge, C.;  D'Hooghe, T.;  De Bie, B.;  Heikinheimo, O.;  Horne, A. W.;  Kiesel, L.;  Nap, A.;  Prentice, A.;  Saridogan, E.;  Soriano, D.; Nelen, W., ESHRE guideline: management of women with endometriosis †. Human Reproduction 2014, 29 (3), 400-412.
  2. Endometriosis and infertility: a committee opinion. Fertility and Sterility 2012, 98 (3), 591-598.
  3. Kuznetsov, L.; Dworzynski, K.;  Davies, M.; Overton, C., Diagnosis and management of endometriosis: summary of NICE guidance. BMJ 2017, 358, j3935.
  4. Pellicer, A.; Oliveira, N.;  Ruiz, A.;  Remohí, J.; Simón, C., Exploring the mechanism(s) of endometriosis-related infertility: an analysis of embryo development and implantation in assisted reproduction. Hum Reprod 1995, 10 Suppl 2, 91-7.
  5. de Ziegler, D.; Pirtea, P.;  Carbonnel, M.;  Poulain, M.;  Cicinelli, E.;  Bulletti, C.;  Kostaras, K.;  Kontopoulos, G.;  Keefe, D.; Ayoubi, J. M., Assisted reproduction in endometriosis. Best Practice & Research Clinical Endocrinology & Metabolism 2019, 33 (1), 47-59.
  6. Nicolaus, K.; Bräuer, D.;  Sczesny, R.;  Jimenez-Cruz, J.;  Bühler, K.;  Hoppe, I.; Runnebaum, I. B., Endometriosis reduces ovarian response in controlled ovarian hyperstimulation independent of AMH, AFC, and women's age measured by follicular output rate (FORT) and number of oocytes retrieved. Arch Gynecol Obstet 2019, 300 (6), 1759-1765.
  7. Mizrachi, Y.; Horowitz, E.;  Farhi, J.;  Raziel, A.; Weissman, A., Ovarian stimulation for freeze-all IVF cycles: a systematic review. Hum Reprod Update 2020, 26 (1), 118-135.
  8. Coccia, M. E.; Rizzello, F.; Gianfranco, S., Does controlled ovarian hyperstimulation in women with a history of endometriosis influence recurrence rate? J Womens Health (Larchmt) 2010, 19 (11), 2063-9.
  9. Chapron, C.; Marcellin, L.;  Borghese, B.; Santulli, P., Rethinking mechanisms, diagnosis and management of endometriosis. Nat Rev Endocrinol 2019, 15 (11), 666-682.
  10. Simopoulou, M.; Rapani, A.;  Grigoriadis, S.;  Pantou, A.;  Tsioulou, P.;  Maziotis, E.;  Tzanakaki, D.;  Triantafyllidou, O.;  Kalampokas, T.;  Siristatidis, C.;  Bakas, P.; Vlahos, N., Getting to Know Endometriosis-Related Infertility Better: A Review on How Endometriosis Affects Oocyte Quality and Embryo Development. Biomedicines 2021, 9 (3).
  11. Dückelmann, A. M.; Taube, E.;  Abesadze, E.;  Chiantera, V.;  Sehouli, J.; Mechsner, S., When and how should peritoneal endometriosis be operated on in order to improve fertility rates and symptoms? The experience and outcomes of nearly 100 cases. Arch Gynecol Obstet 2021.
  12. Abesadze, E.; Sehouli, J.;  Mechsner, S.; Chiantera, V., Possible Role of the Posterior Compartment Peritonectomy, as a Part of the Complex Surgery, Regarding Recurrence Rate, Improvement of Symptoms and Fertility Rate in Patients with Endometriosis, Long-Term Follow-Up. J Minim Invasive Gynecol 2020, 27 (5), 1103-1111.

Reviewer 3 Report

Authors meticulously answered the raised comments and modified the manuscript including more recent evidence from the iterature and checking both linguistic inaccuracies and typing errors.

Moreover, the several changes within the manuscript make the paper more critical and accurate, strengthening its scientific value.

Therefore, the review can now be published in ‘Cells’.

Author Response

Thank you very much